# Congenital Rubella Syndrome Does Not Increase with Introduction of Rubella-Containing Vaccine

**DOI:** 10.3390/vaccines12070811

**Published:** 2024-07-22

**Authors:** Kurt Frey

**Affiliations:** Institute for Disease Modeling, Bill and Melinda Gates Foundation, Seattle, WA 98109, USA; kurt.frey@gatesfoundation.org

**Keywords:** rubella, congenital rubella syndrome, vaccine introduction, agent-based model

## Abstract

Rubella infection is typically mild or asymptomatic except when infection occurs during pregnancy. Infection in early pregnancy can cause miscarriage, stillbirth, or congenital rubella syndrome. Only individuals that are still susceptible to rubella infection during child-bearing age are vulnerable to this burden. Rubella-containing vaccine (RCV) is safe and effective, providing life-long immunity. However, average age-at-infection increases with increasing vaccination coverage, which could potentially lead to increased disease burden if the absolute risk of infection during child-bearing age increases. The dynamics of rubella transmission were explored using EMOD, a software tool for building stochastic, agent-based infection models. Simulations of pre-vaccine, endemic transmission of rubella virus introduced RCV at varying levels of coverage to determine the expected future trajectories of disease burden. Introducing RCV reduces both rubella virus transmission and disease burden for a period of around 15 years. Increased disease burden is only possible more than a decade post-introduction, and only for contexts with persistently high transmission intensity. Low or declining rubella virus transmission intensity is associated with both greater burden without vaccination and greater burden reduction with vaccination. The risk of resurgent burden due to incomplete vaccination only exists for locations with persistently high infectivity, high connectivity, and high fertility. A trade-off between the risk of a small, future burden increase versus a large, immediate burden decrease strongly favors RCV introduction.

## 1. Introduction

Infection by the rubella virus is typically mild [1], with symptoms that are often sub-clinical or absent [2]. However, infection in early pregnancy can cause miscarriage, stillbirth, or congenital rubella syndrome (CRS) [3]. Only individuals that are still susceptible to rubella infection during child-bearing age are vulnerable to this burden.

A single dose of rubella-containing vaccine (RCV) induces high seroconversion rates (≥95%) and provides long-term immunity, like that produced by natural infection. In most countries, RCV is administered along with the measles vaccine and follows the two-dose schedule for measles. Rubella control efforts benefit from established measles control programs; RCV introduction is typically achieved by switching from monovalent measles-containing vaccine (MCV) to bivalent measles–rubella (MR) vaccine. This switch leverages established delivery modes and avoids many logistical challenges associated with new vaccine introduction. Switching to a bivalent vaccine leads to around a two-fold increase in its cost, but the vaccine cost is a minority component in the total cost of delivery [4].

The global incidence of rubella and the burden of CRS has decreased significantly since the introduction of a rubella-containing vaccine in 1969. A majority of the remaining burden is in countries that do not routinely vaccinate for rubella [5,6]. At the end of 2023, nineteen countries had yet to introduce RCV into their recommended schedule for vaccination. As shown in Figure 1, these countries are predominantly in sub-Saharan Africa. While the economic benefits of rubella vaccination are large [7], resource constraints in low- and lower-middle-income countries are also substantial. Operational considerations may limit the reach of health services and result in incomplete vaccination, which has the potential to lead to an increase in disease burden [8].

The average age-at-infection increases with increasing vaccination coverage, which has the potential to increase the disease burden if the absolute risk of rubella infection during child-bearing age increases. The primary consideration when deciding to introduce RCV is ensuring that the burden of disease will go down. Current guidance from the World Health Organization suggests that countries planning to introduce RCV should have at least
80% coverage with the first dose of the measles vaccine [10]. This level of coverage was established to support the control of rubella in very-high-transmission settings [11] and is sufficient for elimination in nearly all contexts.

Sustained rubella transmission tends to occur only in large populations, with interruption of transmission occurring during epidemic troughs in smaller communities [12]. Local extinction will tend to increase average age-at-infection by allowing individuals to remain susceptible for longer, possibly extending into child-bearing age. Outbreaks following re-introduction of the pathogen have the potential for elevated burden because of this accumulation of susceptibility.

Any approach to vaccination that ensures immunity before child-bearing age will lead to a reduced burden. Historically, two primary approaches were considered: selective vaccination of girls and young women, and universal vaccination of infants [8]. Selective vaccination aims to reduce burden without interrupting transmission, but can still result in an accumulation of susceptibility leading to severe outbreaks [13]. Universal vaccination leading to the interruption of transmission is the preferred strategy for controlling the rubella burden [10], although selective vaccination can be a useful supplement when infant vaccination rates are low.

After RCV introduction, population immunity for rubella continues to be supported through supplemental immunization activities (SIAs), although the schedule for these activities is dictated by needs for measles control [14].

This study explores the dynamics of rubella virus transmission and rubella burden following RCV introduction at various levels of coverage. Simulations explore the implications of declining fertility (an expected part of demographic transition), incomplete population mixing, and heterogeneous infectivity. Constant birth rates, well-mixed connectivity, and homogeneous infectivity are all useful approximations for modeling pathogen transmission in a population, but will have a tendency to bias outcomes. This bias may result in incorrect, unnecessarily conservative policy guidance.

## 2. Materials and Methods

The dynamics of rubella transmission were explored using EMOD, a software tool for building stochastic, agent-based infection models [15]. Details of the rubella model have been described previously [16]. That model has been generalized by applying population, fertility, and mortality estimates for sub-Saharan Africa from the United Nations World Population Prospects (WPP) [17].

New agents are added on each time step, and start as either maternally protected or susceptible. Mortality is possible for all conditions and is unrelated to infection. Vaccination that occurs while an agent is maternally protected, exposed, or infected is assumed to not provide immunity. Maternal protection ends after a mean duration of three months. Immunity to rubella was assumed to be binary (i.e., either present or absent). Immunity from infection or vaccination was assumed to provide life-long protection and did not wane.

The simulations are initialized in the year 2000 starting with a population of 1 million; outcomes from the first 20 years are ignored as endemic pathogen transmission is established. The results for the subsequent 40 years, from 2020 to 2060, are depicted for a variety of scenarios. In each scenario, RCV is introduced in 2025 at several levels of coverage. All scenarios include
0% coverage as a reference to illustrate continued endemic transmission without vaccination. The use of a stochastic, agent-based model allows for exploring the dynamics of interruption and re-introduction.

The rubella burden includes miscarriage, stillbirth, and CRS. This study groups those outcomes together. The likelihood of burden associated with a rubella infection was approximated as
80% if infection occurs during the first trimester of pregnancy, and
50% if infection occurs in the four weeks prior to conception or in the first four weeks of the second trimester of pregnancy [18,19]. The estimated burden scales linearly with these probabilities, which multiply age-structured infection rates and age-structured fertility rates. The relative burden with respect to baseline scenarios without vaccination is independent of these probabilities.

Infectivity has been assumed to be uniform over all age groups. Other investigators [6,20,21] consider age-structure in the force of infection in order to more closely match the observed serological profiles for rubella seronegativity (e.g., <13 years, and 13+ years; <3 years, 3–15 years, and 15+ years). Uniform infectivity was selected as the best representation for about half of the countries examined [6]. Data quality is a significant consideration in those estimates, and a moderate reduction in infectivity for older age groups seems to be the most likely case. Uniform infectivity is a simplifying assumption as well as a conservative one from the perspective of risk associated with incomplete vaccination. Reduced infectivity in older cohorts would lead to lower rubella burden and lower risk of resurgent burden.

## 3. Results

The modeled results depict a mean rate of infection and burden for the various scenarios; each trajectory is averaged over ten-thousand stochastic outcomes. Individual timeseries are highly variable and include both outbreaks and periods of elimination. Persistent importation of rubella virus ensures that elimination only occurs when simulated immunity levels exceed the threshold for herd immunity (i.e.,
$1-1/R_{0}$). Above this threshold, infections still occur because of importations, but they do not lead to sustained chains of transmission.

### 3.1. Representative Timeseries for Rubella Burden

The outcomes depicted in Figure 2 are of RCV introduction in 2025 through the routine immunization system only. The recommended vaccine introduction strategy is to simultaneously implement a wide age range (9 months–15 years) vaccination campaign as well [10]. That approach leads to a step-change decrease in both infections and burden, with probable (although potentially impermanent) elimination of rubella. The catch-up campaign has been omitted in these scenarios to more clearly illustrate the dynamics of infection and burden. The age structure of the population is held constant at 2020 levels so that clearly defined equilibrium rates exist.

Following vaccine introduction, a new equilibrium infection rate is established within a period of about 5 years. However, the rate of disease burden equilibrates over a period of 15 to 20 years. Time scales for infection are more rapid because they depend only on virus transmission. The burden of rubella infection depends on age-structured susceptibility, which is a demographic process that changes more slowly.

All rates of vaccination initially reduce burden. The absence of a wide age range vaccine introduction campaign, or mediocre rates of vaccination cannot lead to an immediate burden increase. The potential for increased burden exists only around 15 years post-vaccine introduction as a new, potentially susceptible cohort of individuals reaches child-bearing age.

The average timeseries depicted in Figure 2 incorporate large and potentially increasing levels of variance. As vaccination coverage increases, the variability of outbreaks also increases [22]. For simulations with steady-state populations, the average burden levels over the period 2050 to 2060 are near their long-term equilibrium values. Histograms of annual burden during this decade for the no vaccination and
60% vaccination scenarios are shown in Figure 3.

Although the mean value of both histograms in Figure 3 is around
2.2 per thousand births per year, the median annual burden in scenarios with
60% vaccination is lower, and the
95% bounds on the annual burden span a larger interval: no vaccination
=(0.3,4.7) and
60% vaccination
=(0.0,8.2). The interruption of transmission becomes more likely in scenarios with vaccination, but persistent elimination is not possible because of pathogen importation.

### 3.2. Impact of a Demographic Transition

The outcomes depicted in Figure 4 reproduce those in Figure 2, but do not enforce a constant population pyramid. The age structure of the population evolves as described by medium forward projections from the UN WPP for sub-Saharan Africa. Declining fertility affects infection rates in a manner that is analogous to moderately increasing rates of vaccination.

Absent vaccination, rubella incidence will trend downward, and burden will trend upward. Any non-zero rate of RCV use will lead to greater reductions in burden than would be estimated when assuming a steady-state population structure.

### 3.3. Population Connectivity Considerations

The outcomes depicted in Figure 5 reproduce those in Figure 2, but subdivide the population into four non-interacting, but otherwise identical sub-populations. The total simulated population remains unchanged, and the rate of pathogen importation events also remains unchanged.

The equilibrium infection rates are similar to simulations with a well-mixed population, although the time scale for attaining that equilibrium increased. The birth rates and vaccination rates are the same in both scenarios; outbreaks are limited by the size of the sub-population, leading to overall slower dynamics and an accumulation of susceptible agents. The equilibrium annual burden is elevated, around
3.0 per thousand births, compared to
2.2 in well-connected simulations, which aligns with the accumulation of susceptibility and an increased average age-at-infection.

### 3.4. Variability of Transmission Intensity

Assumptions regarding the intensity of pathogen transmission have the largest effect on the estimated rubella burden. The simulations examining demographics and connectivity assumed a transmission intensity such that the mean
$R_{0}$ value was
5.0, near the middle of the typical range of
3.0 to
8.0 [23]. The outcomes when assuming
$R_{0}$ is
3.0 are presented in Figure 6; the outcomes when assuming
$R_{0}$ is
7.0 are presented in Figure 7.

The infection rates absent vaccination are roughly equivalent in both scenarios. Rubella infection provides lifelong immunity, and almost all susceptible individuals will eventually be infected, even at lower transmission intensity. However, an individual’s expected age-at-infection increases in the low transmission scenarios, leading to an elevated burden. This effect is non-linear over the range illustrated. Increasing
$R_{0}$ from
5.0 to
7.0 halves the burden in the no-vaccine scenarios, while reducing
$R_{0}$ from
5.0 to
3.0 nearly doubles the burden.

At lower transmission intensity, vaccination decreases the rubella burden relative to baseline at all levels of coverage. At higher transmission intensities, intermediate levels of vaccination have elevated levels of equilibrium burden, and higher coverage is required to ensure a decrease in burden.

## 4. Discussion

The outcomes are presented using mean trajectories, which do not highlight important differences in the distribution of burden. The likelihood of interruption increases with increasing vaccination, with more frequent periods of local extinction and zero infection. Endemic transmission at low vaccination coverages transitions into long periods of interruption, where importation can lead to a large outbreak. At intermediate vaccination coverages, where the long-term average rubella burden may appear comparable to scenarios without vaccination, the median burden will still have decreased. With increasing vaccination coverage, the total burden is concentrated into fewer, larger outbreaks.

The omission of a wide age range catch-up campaign in these examples is only to emphasize the disease dynamics. The introduction of RCV without a campaign is associated with a potential resurgence in rubella burden [24,25]. The introduction of RCV initially leads to declining incidence across all age groups, but many individuals <15 years old at the time of introduction are still susceptible and would be ineligible for vaccination through routine services. Accumulation of susceptibility increases outbreak risk, and vaccinating these individuals at the time of RCV introduction via a wide age range campaign should be a priority.

The baseline scenarios depict a setting with
$R_{0}=5.0$, where a
60% rate of vaccination is sufficient to avoid a long-term increase in the average burden. These scenarios were selected for demonstration purposes. Not all levels of rubella infectivity are equally likely, and values
$R_{0}<5.0$ are more probable [23,26].

Accurately incorporating the expected demographic transition into baseline scenarios reduces the rate of vaccination that avoids a long-term increase in the average burden to
50%. All simulations leverage demographics for sub-Saharan Africa, which are generally appropriate for countries that have not yet introduced the vaccine, but are not specific to any individual country. Above average fertility rates would be expected to decline more rapidly, and below average rates would be expected to decline more slowly. Both variations, either fertility declining more rapidly than average, or starting at a lower level than average, reduce the likelihood of increased burden due to incomplete vaccination.

Disconnected sub-populations result in slower dynamics than the well-mixed populations in the baseline scenario. These outcomes emphasize the value of implementing wide-age range vaccination campaigns concurrent to vaccine introduction through the routine immunization system. Vaccination campaigns lead to step-change decreases in both infection rate and burden, which provide the greatest benefit to disconnected sub-populations, such as those that may be geographically remote or rural locations. Reduced connectivity also elevates burden with respect to the baseline scenario, which again corresponds to greater reductions in burden when vaccinating.

Rubella infectivity is uncertain for most settings, and sub-national variation is expected in large, heterogeneous countries, such as Nigeria [21] or the Democratic Republic of the Congo [16]. Neglecting demographic effects and connectivity effects, the demonstration outcomes show a vaccination rate of
60% would decrease the equilibrium rate of burden for
$R_{0}=3.0$ (from 4 to 2), increase the equilibrium rate of burden for
$R_{0}=7.0$ (from 1 to 2), and not change the equilibrium rate for
$R_{0}=5.0$. If all levels of infectivity were equally likely, the average outcome for these three scenarios would be to reduce the overall burden. That result contrasts with the outcome for the average scenario (
$R_{0}=5.0$), which is no reduction in burden. Heterogeneity in infectivity leads to greater burden reduction when vaccinating.

The outcomes presented here are not intended to identify appropriate vaccination thresholds for RCV introduction; they emphasize that a fixed, generally applicable threshold is inappropriate. The risk for increased rubella burden associated with incomplete vaccination occurs decades in the future and only for settings with persistently high fertility, high connectivity, and high infectivity. It is not a hypothetical risk, but it is especially unlikely. A focus on mitigating that risk accepts the certainty of increasing rubella burden absent vaccination.

This study is not specific to any particular country. It uses demographics that are an average for sub-Saharan Africa and infectivity (
$R_{0}=5.0$) that is also an average for the region [27]. Each country that has not yet introduced RCV will have individual needs that are not average. However, all countries do feature the same overall trends. All countries have declining birth rates, all have regions that are less than completely well mixed, and all have some degree of heterogeneity in infectivity.

After RCV introduction, ongoing control efforts will be driven by measles considerations because of that virus’ greater infectiousness and the combined formulation of the vaccine. Declining overall rubella incidence and a tendency toward greater variability in that incidence will require increased emphasis on surveillance to identify any sporadic transmission and ensure that it does not lead to an outbreak of disease.

## 5. Conclusions

The guidance that
80% coverage of measles-containing vaccine is a necessary prerequisite for rubella vaccine introduction was created to ensure against any possibility of a future resurgence in burden. That guidance is unnecessarily conservative. It does not account for the continuously increasing current rate of burden that exists absent vaccination, which is a consequence of falling birth rates.

Rubella vaccine introduction supported by a wide age range vaccination campaign will lead to a large decrease in burden for a period of at least 15 years. The risk of a future burden resurgence due to incomplete vaccination only exists for communities with persistently high infectivity, high connectivity, and high fertility. After vaccine introduction, an emphasis on surveillance will help identify any sporadic transmission and support communities that may be at risk of outbreaks.

## Figures and Tables

**Figure 1 vaccines-12-00811-f001:**
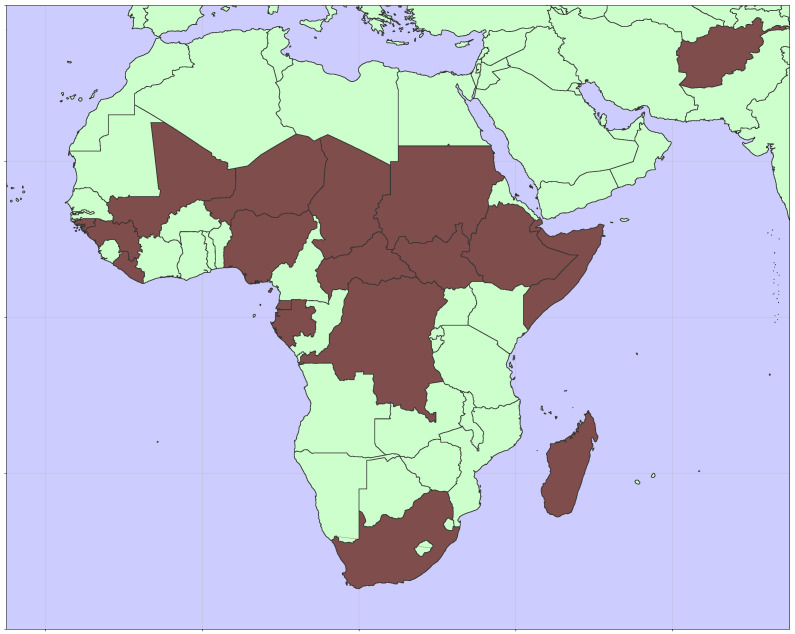
Map showing the locations of nineteen countries yet to introduce RCV [9].

**Figure 2 vaccines-12-00811-f002:**
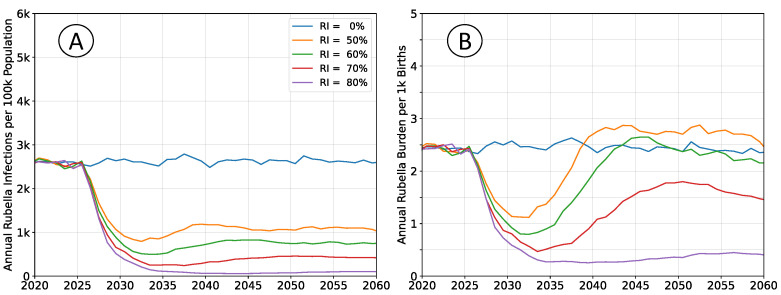
Average simulation outcomes for a steady-state population. (**A**) Mean annual incidence of rubella virus infections per 100 k total population as a function of time following RCV introduction through routine immunization only. (**B**) Mean annual burden of rubella virus infection per 1 k births.

**Figure 3 vaccines-12-00811-f003:**
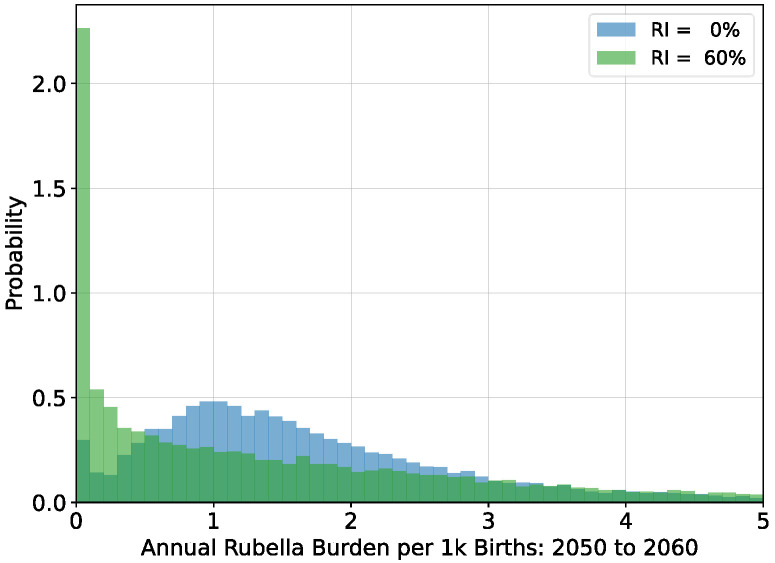
Histograms of annual rubella burden estimated over the period 2050 to 2060 for simulations of steady-state populations with no vaccination and with
60% vaccination. Average burden for both distributions is around 2.2 per thousand births per year.

**Figure 4 vaccines-12-00811-f004:**
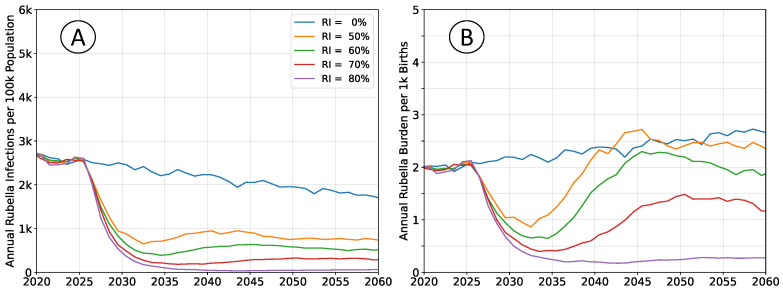
Average simulation outcomes incorporating a changing population structure. (**A**) Mean annual incidence of rubella virus infections per 100 k total population as a function of time following RCV introduction through routine immunization only. (**B**) Mean annual burden of rubella virus infection per 1 k births.

**Figure 5 vaccines-12-00811-f005:**
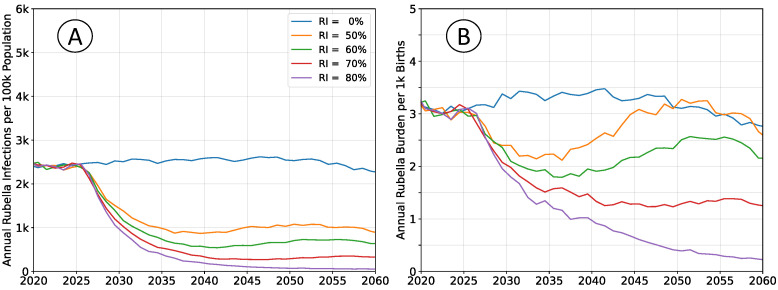
Average simulation outcomes for a steady-state population, divided into multiple non-interacting sub-populations. (**A**) Mean annual incidence of rubella virus infections per 100 k total population as a function of time following RCV introduction through routine immunization only. (**B**) Mean annual burden of rubella virus infection per 1 k births.

**Figure 6 vaccines-12-00811-f006:**
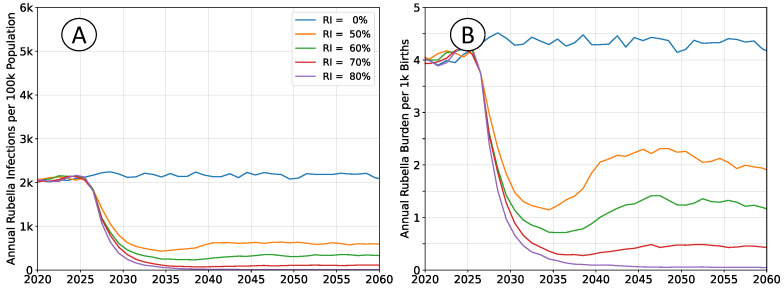
Average simulation outcomes assuming
$R_{0}=3.0$. (**A**) Mean annual incidence of rubella virus infections per 100 k total population as a function of time following RCV introduction through routine immunization only. (**B**) Mean annual burden of rubella virus infection per 1 k births.

**Figure 7 vaccines-12-00811-f007:**
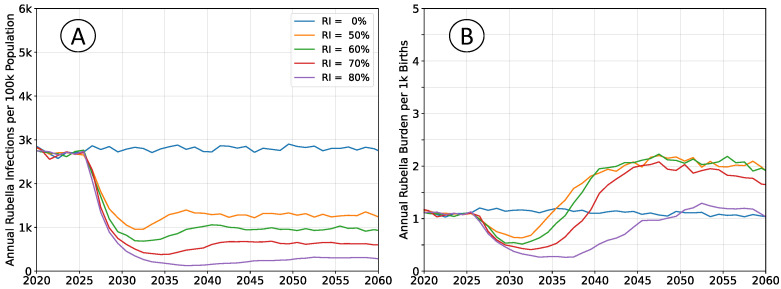
Average simulation outcomes assuming
$R_{0}=7.0$. (**A**) Mean annual incidence of rubella virus infections per 100 k total population as a function of time following RCV introduction through routine immunization only. (**B**) Mean annual burden of rubella virus infection per 1 k births.

## Data Availability

Source code for EMOD is available on GitHub: https://github.com/InstituteforDiseaseModeling/EMOD-Generic, accessed on 1 July 2024. Implementation of the rubella model and configurations for the simulations presented here are hosted in a separate repository: https://github.com/InstituteforDiseaseModeling/EMOD-Generic-Scripts, accessed on 1 July 2024.

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
