# Peer review of "Congenital Rubella Syndrome Does Not Increase with Introduction of Rubella-Containing Vaccine"

_vaccines, 2024, doi:10.3390/vaccines12070811_

Round 1
Reviewer 1 Report
Comments and Suggestions for Authors
The authors have used the EMOD software tool for building stochastic, agent-based infection models, and projected the effect of Rubella Containing Vaccine for the next 40 years. The projection has been analyzed for changing population structure, predicting an increase in disease transmission with increasing maternal age, thereby suggesting a combination of selective vaccination through a supplementalimmunization activity, of girls and young women and universal vaccination of infants. Graphical projection of the data helps to depict their observation.
This article will help formulate policies for Rubella vaccination programs.
Suggestion:
1. A few lines on reasons and issues in the 19 countries that are yet to introduce universal rubella vaccination for infants will help to address problems with implementing the program in these countries.
2. The discussion could be improved by highlighting the implications of their study for establishing policies by countries to improve the rubella immunization program
Author Response
The authors have used the EMOD software tool for building stochastic, agent-based infection models, and projected the effect of Rubella Containing Vaccine for the next 40 years. The projection has been analyzed for changing population structure, predicting an increase in disease transmission with increasing maternal age, thereby suggesting a combination of selective vaccination through a supplemental immunization activity, of girls and young women and universal vaccination of infants. Graphical projection of the data helps to depict their observation.
This article will help formulate policies for Rubella vaccination programs.
Comment 1:
A few lines on reasons and issues in the 19 countries that are yet to introduce universal rubella vaccination for infants will help to address problems with implementing the program in these countries.
Thank you for the suggestion, I have added additional background around considerations for rubella introduction.
Comment 2:
The discussion could be improved by highlighting the implications of their study for establishing policies by countries to improve the rubella immunization program
I have expanded the discussion to include policy implication for rubella programs.
Reviewer 2 Report
Comments and Suggestions for Authors
The manuscript explores the potential impact of introducing the Rubella Containing Vaccine (RCV) on the burden of Congenital Rubella Syndrome (CRS). This study is important since it addresses both the benefits and potential long-term risks associated with RCV introduction.
Comments:
I suggest adding a statement in the conclusion emphasizing the need for a wide age range campaign while ensuring high vaccination coverage to prevent any potential increase in CRS burden.
The assumption regarding uniform infectivity across all age groups could be further validated with real-world data and support the conclusions.
Comments on the Quality of English Language
No specific comments
Author Response
Comment 1:
I suggest adding a statement in the conclusion emphasizing the need for a wide age range campaign while ensuring high vaccination coverage to prevent any potential increase in CRS burden.
Thank you for the suggestion, I have added additional discussion emphasizing the importance of a wide age range catch-up at the time of vaccine introduction.
Comment 2:
The assumption regarding uniform infectivity across all age groups could be further validated with real-world data and support the conclusions.
Uniform infectivity across all age groups is an assumption with mixed support in the literature. One of the most comprehensive analyses is from Vynnycky et al. (https://journals.plos.org/plosone/article?id=10.1371/journal.pone.0149160). In that work, infectivity was estimated to be uniform across ages in about half of the data sets examined. Those estimates are largely a consequence of data quality, and a moderate reduction in infectivity within older age groups seems to be the most likely case.
In this work, uniform infectivity was assumed because it is associated with a greater risk of resurgent burden and the intent is to characterize that burden risk as small even with conservative assumptions.
I have updated the manuscript to clarify the motivation for the assumption of uniform infectivity.
Reviewer 3 Report
Comments and Suggestions for Authors
Estimated Editors in "Vaccines",
I'v been invited to provide a review of the paper entitled "Congenital Rubella Syndrome Burden is Unlikely to Increase with Introduction of Rubella Containing Vaccine" from prof. Frey of Gates Foundation.
In this study, prof. Frey assesses the potential impact of massive RCV implementation in Subsaharan countries. The Author provides several estimates according to the potential vaccination rates, and readers are warranted about potential costs/benefits impact of the improved vaccination policies.
From my point of view, the paper is concise and well written, but it is affected by some potential flaws.
At least from the main text in its current stage of development, the model was a cumulative one. But Subsaharan Africa is a very diverse context, with countries experiencing a diachronous development in terms of population and economic settings. I'm guessing whether a meta-analytical approach would be more appropriate, by providing single-country estimates and also estimates on the heterogeneity of estimates. Internal heterogeneity from each sampled countries (e.g. Congo Democratic Republic, with "core" areas around the Capital city vs. areas that have been involved in conflicts and where the central government has a very limited control) would be quite difficult to address and manage, but could be discussed in the final section of the paper.
Moreover, also the RI estimates (see figure 2-6) have been provided as point values, while a more cautious approach would provide 95%CI for all analyses.
Author Response
Comment 1:
At least from the main text in its current stage of development, the model was a cumulative one. But sub-Saharan Africa is a very diverse context, with countries experiencing a diachronous development in terms of population and economic settings. I'm guessing whether a meta-analytical approach would be more appropriate, by providing single-country estimates and also estimates on the heterogeneity of estimates. Internal heterogeneity from each sampled countries (e.g. Congo Democratic Republic, with "core" areas around the Capital city vs. areas that have been involved in conflicts and where the central government has a very limited control) would be quite difficult to address and manage, but could be discussed in the final section of the paper.
Thank you for the feedback. Single country estimates have been the focus of VIMC models (https://www.vaccineimpact.org/rubella), and current work by those groups is developing the analysis you describe. The intention of this work is to not duplicate that effort, and instead focus on generalized learnings that are applicable in every setting. Declining birth rate, variable connectivity, and heterogeneity of infectivity lead to reduced risk from incomplete vaccination for all countries.
Subnational heterogeneity in infectivity is especially significant for larger nations like Nigeria and the DRC, and detailed subnational analyses have been published for those countries. Example simulations at R0 values of 3, 5, and 7 illustrate that the risk due incomplete vaccination averaged across those three outcomes is less than the risk estimated at the averaged outcome (R0 = 5). Subnational heterogeneity leads to lower overall risk, but can also concentrate that risk in fewer areas. I have expanded the discussion on this point for clarity and emphasis.
Comment 2:
The RI estimates (see figure 2-6) have been provided as point values, while a more cautious approach would provide 95%CI for all analyses.
Variance in the outcomes presented in figures 2 - 6 is large. Higher rates of vaccination are also associated with higher variability in outcome, so the shape of the distribution of outcomes is also different at each level of vaccination. I have added an additional figure (now figure 3) that profiles the uncertainty and distribution of outcomes associated with the base case simulations presented in figure 2. Variance in outcomes for the other scenarios is analogous.
Round 2
Reviewer 3 Report
Comments and Suggestions for Authors
The authors has either addressed or solved the reasons of my concerns; as a consequence I've no further requests.